# Comparative Evaluation of Large Language and Multimodal Models in Detecting Spinal Stabilization Systems on X-Ray Images

**DOI:** 10.3390/jcm14103282

**Published:** 2025-05-08

**Authors:** Bartosz Polis, Agnieszka Zawadzka-Fabijan, Robert Fabijan, Róża Kosińska, Emilia Nowosławska, Artur Fabijan

**Affiliations:** 1Department of Neurosurgery, Polish-Mother’s Memorial Hospital Research Institute, 93-338 Lodz, Poland; jezza@post.pl (B.P.); roza.w.kosinska@gmail.com (R.K.); emilia.nowoslawska@iczmp.edu.pl (E.N.); 2Department of Rehabilitation Medicine, Faculty of Health Sciences, Medical University of Lodz, 90-419 Lodz, Poland; agnieszka.zawadzka@umed.lodz.pl; 3Independent Researcher, Luton LU2 0GS, UK; robert.f.fabijan@gmail.com

**Keywords:** scoliosis, artificial intelligence, BiomedCLIP, ChatGPT-4o, medical image analysis, machine learning in healthcare

## Abstract

**Background/Objectives:** Open-source AI models are increasingly applied in medical imaging, yet their effectiveness in detecting and classifying spinal stabilization systems remains underexplored. This study compares ChatGPT-4o (a large language model) and BiomedCLIP (a multimodal model) in their analysis of posturographic X-ray images (AP projection) to assess their accuracy in identifying the presence, type (growing vs. non-growing), and specific system (MCGR vs. PSF). **Methods:** A dataset of 270 X-ray images (93 without stabilization, 80 with MCGR, and 97 with PSF) was analyzed manually by neurosurgeons and evaluated using a three-stage AI-based questioning approach. Performance was assessed via classification accuracy, Gwet’s Agreement Coefficient (AC1) for inter-rater reliability, and a two-tailed z-test for statistical significance (*p* < 0.05). **Results:** The results indicate that GPT-4o demonstrates high accuracy in detecting spinal stabilization systems, achieving near-perfect recognition (97–100%) for the presence or absence of stabilization. However, its consistency is reduced when distinguishing complex growing-rod (MCGR) configurations, with agreement scores dropping significantly (AC1 = 0.32–0.50). In contrast, BiomedCLIP displays greater response consistency (AC1 = 1.00) but struggles with detailed classification, particularly in recognizing PSF (11% accuracy) and MCGR (4.16% accuracy). Sensitivity analysis revealed GPT-4o’s superior stability in hierarchical classification tasks, while BiomedCLIP excelled in binary detection but showed performance deterioration as the classification complexity increased. **Conclusions:** These findings highlight GPT-4o’s robustness in clinical AI-assisted diagnostics, particularly for detailed differentiation of spinal stabilization systems, whereas BiomedCLIP’s precision may require further optimization to enhance its applicability in complex radiographic evaluations.

## 1. Introduction

Open-source artificial intelligence models (OSAIM) are freely available, publicly accessible tools that have found extensive applications in both computer science and medicine, significantly contributing to advancements in diagnostics and treatment planning [1,2,3,4,5,6]. While these models offer advanced capabilities for processing and interpreting visual data, most are not specifically designed for the analysis of medical images, such as X-ray scans. To address this gap, specialized models like BiomedCLIP have been developed, integrating natural language processing with medical image analysis, thereby opening new possibilities for diagnostic support and treatment selection [7]. Increasing research on the clinical applications of OSAIM underscores their potential in radiology and other fields of medicine.

In the past year, there has been a growing global interest in open-source AI models, exemplified by the development of advanced language models such as ChatGPT by OpenAI. One of the key products of this organization is Contrastive Language–Image Pretraining (CLIP), which enables zero-shot image classification, image captioning, and visual question answering [8]. Trained on extensive image-text datasets, CLIP serves as the foundation for specialized models such as SDA-CLIP, designed for surgical activity analysis, and SleepCLIP, which supports the diagnosis of sleep disorders [9,10]. One of the most advanced CLIP-based models is BiomedCLIP (Bio-med-CLIP-PubMedBERT_256-vit_base_patch16_224), which integrates natural language processing with medical image interpretation. Trained on the PMC-15M dataset, comprising 15 million image−caption pairs from PubMed Central, BiomedCLIP utilizes PubMedBERT for medical text analysis and Vision Transformer for biomedical imaging. As a result, BiomedCLIP demonstrates high efficacy in image classification, cross-modal retrieval, and question−answering, making it a valuable tool for diagnostics, research, and medical education [11].

In our previous study, we evaluated BiomedCLIP in detecting and classifying scoliosis in pediatric posturographic X-ray images. The analysis revealed that the model performed well in identifying advanced cases of scoliosis, particularly in severe stages. However, its accuracy was limited when detecting milder forms and distinguishing between single-curve and double-curve scoliosis. These findings highlight BiomedCLIP’s potential in medical image analysis, while also indicating the need for further refinement of the model and expansion of the training dataset to enhance its clinical applicability [7].

ChatGPT, as an advanced language model, has also been explored for medical diagnostics and biomedical data analysis, although its effectiveness remains under investigation. Suthar et al. reported that ChatGPT-4 achieved a diagnostic accuracy of 57.86%, highlighting both its potential and limitations [12]. Lee et al. compared its ability to analyze chest X-ray images with the KARA-CXR algorithm, demonstrating that dedicated AI systems outperformed ChatGPT in medical image interpretation [13]. Additionally, Sohail examined ChatGPT’s early impact on medical sciences and biomedical engineering, emphasizing both its diagnostic potential and the challenges related to accuracy and interpretability [14].

In our studies, we assessed the ability of AI models, including ChatGPT and Microsoft Bing, to diagnose single-curve scoliosis from posturographic X-ray images. ChatGPT demonstrated 100% accuracy in scoliosis detection, but its Cobb angle assessment accuracy was only 43.5%, showing significant discrepancies compared to expert evaluations. Moreover, the model struggled with determining curve direction, scoliosis classification, and vertebral rotation detection, while Microsoft Bing failed to detect scoliosis in any case. These findings suggest that while AI holds promise in scoliosis identification, significant improvements in image analysis and precise diagnostic assessment are required before these models can be reliably implemented in clinical practice [15].

Scoliosis is defined as a three-dimensional spinal deformity in which the coronal plane curvature exceeds 10 degrees, measured using the Cobb method, which serves as the clinical diagnostic standard [16]. The gold standard for scoliosis diagnosis is X-ray imaging of the spine in two projections: anteroposterior (AP) and lateral. This imaging approach enables precise assessment of spinal curvature severity, vertebral rotation, and deformity progression, facilitating the planning of appropriate therapeutic procedures [17,18]. Scoliosis can be classified based on various criteria, including patient age, etiology, curve location, and severity measured by the Cobb angle [19].

The management of scoliosis is highly individualized but generally follows established clinical guidelines. According to AO Spine recommendations, mild scoliosis (10–20 degrees) is typically managed through regular monitoring and physiotherapy, while moderate scoliosis (20–40 degrees) may require orthotic bracing. In severe cases (>40 degrees), surgical intervention is often indicated [19]. In adolescent idiopathic scoliosis (AIS), the most common form of spinal deformity in children, treatment options range from conservative therapy (e.g., physiotherapy) to surgical interventions, such as dynamic fixation or posterior spinal fusion (PSF) [20].

For early-onset scoliosis (EOS), occurring before the age of 10, growth-friendly surgical techniques such as magnetically controlled growing rods (MCGR) are frequently used to facilitate spinal and thoracic development while maintaining deformity correction. Traditional approaches, such as posterior spinal fusion (PSF), may lead to complications, including thoracic insufficiency syndrome, growth restriction, and respiratory issues, if performed too early. MCGR systems offer a promising alternative by reducing the need for repeated surgical lengthening procedures, thereby minimizing complications and psychosocial burden on young patients [21] (Figure 1).

The primary objective of this study is to compare two types of artificial intelligence models—large language models (LLLM), such as ChatGPT-4o (Omni), and the multimodal model BiomedCLIP—in terms of their ability to detect and classify spinal stabilization systems in posturographic X-ray images in the anteroposterior (AP) projection.

### 1.1. Justification

The selection of LLLM (ChatGPT-4o) and BiomedCLIP for comparison is based on their differing approaches to data analysis and their specific applications in medical imaging diagnostics:ChatGPT-4o:
○Although primarily a language model, ChatGPT-4o demonstrates strong capabilities in image analysis, particularly in recognizing detailed objects and structures.○With its image-processing functionality, it can be applied to detect complex structures, such as spinal stabilization systems, especially when they are clearly visible on X-ray images.○Its primary advantage lies in its ability to generate precise descriptions and interpret results, making it valuable in medical applications where detailed descriptions are essential.
BiomedCLIP:
○BiomedCLIP is a multimodal model that integrates images and text, enabling it to compare visual features of an image with textual descriptions.○Its performance depends on the quality of the provided textual descriptions and the accuracy of image−text associations.○This model is particularly useful for tasks where textual context aids in recognizing and classifying objects in medical images.○However, its efficacy may be limited in more complex scenarios, such as distinguishing subtle details in medical imaging.

This study aims to determine which of these models is more suitable for medical diagnostics, particularly in the radiographic assessment of spinal stabilization systems in surgical applications. A comparative analysis of these two AI approaches can provide valuable insights into the effectiveness of artificial intelligence in medicine, especially in automating diagnostic processes.

### 1.2. Research Hypotheses


*Accuracy in detecting spinal stabilization systems:*


**H1.** 
*BiomedCLIP will demonstrate significantly higher accuracy in detecting the presence of spinal stabilization systems on posturographic anterior–posterior (AP) X-ray images compared to ChatGPT-4o.*



*Consistency of model responses:*


**H2.** 
*BiomedCLIP will exhibit greater consistency (i.e., reproducibility of results) in classifying X-ray images regarding the presence and type of spinal stabilization systems than ChatGPT-4o.*



*Effectiveness in classifying the type and nature of the stabilization system:*


**H3.** 
*While ChatGPT-4o may be more effective in recognizing the general presence of spinal stabilization systems, BiomedCLIP will outperform in the precise classification of the system type (growing vs. non-growing) and specific system (MCGR vs. PSF).*


## 2. Materials and Methods

The study was conducted at the Polish Mother’s Memorial Hospital Research Institute as part of scientific research activities. The bioethics committee determined that the analysis of the obtained radiological images did not require formal approval. This study, conducted between January 2024 and December 2024, focused on analyzing radiological images in the anterior–posterior (AP) projection in patients aged 3 to 17 years. The X-ray images were collected from January 2022 to December 2024, and out of 590 available images, a total of 270 images were selected, including:93 posturographic AP images without a visible stabilization system,80 images with a visible growing rod system (MCGR),97 images showing the final stabilization (PSF).

Consent for the use of X-ray images was obtained from the legal guardians of the patients. In accordance with data protection principles, all personal information was anonymized.


*Inclusion and Exclusion Criteria:*


The inclusion criteria required technically correct images. For images without a stabilization system, only cases of severe scoliosis (>40° Cobb angle) were included, in line with AO Spine guidelines, which qualify curves exceeding 40° for surgical correction. Additionally, images with a visible MCGR system and final stabilization (PSF) were selected. Image quality was assessed to exclude illegible images, errors in image stitching, or improper framing.


*The exclusion criteria included:*


Images that were improperly stitched,X-rays covering an incomplete view of the spine,Scoliosis cases with additional bone defects, such as hyperkyphosis,Images featuring additional implants unrelated to the studied stabilization systems.


*Imaging and Analysis:*


All of the tests were performed using the same radiological equipment, and the X-ray images were not modified in any way. They were saved in JPEG format with a resolution of 2663 × 1277 px.

The dataset was evaluated by two independent neurosurgery specialists (B.P. and E.N.), who focused on 270 AP posturographic images, including 93 cases of severe scoliosis, with curvatures ranging from 45° to 96°. The analysis aimed to develop baseline scoliosis descriptions, detailing key parameters such as:Degree of deformation (Cobb method),Precise identification of the affected spine segment.

These descriptions were later used to evaluate the capabilities and classification accuracy of open AI systems in scoliosis assessment.

### 2.1. Manual Measurement

The evaluation of posturographic X-ray images was carried out independently by two neurosurgery specialists. For image analysis and Cobb angle measurements, RadiAnt software (version 2023.1) was utilized.

### 2.2. AI System Evaluation Methodology

As part of this study, an analysis of artificial intelligence models was conducted, including ChatGPT-4o (OpenAI, San Francisco, CA, USA). The experiments took place between 1 January and 30 January 2025, following a three-stage methodology designed to assess the model’s ability to interpret posturographic X-ray images.


*Three-Stage Questioning Approach*


1.General Image Interpretation

Question: ‘*What do you see in the image?*’

This open-ended question allowed the model to describe the X-ray without any prior hints, such as the presence or absence of a spinal stabilization system.

2.Detection of a Stabilization System

Question: ‘*Do you observe any spinal stabilization system in the image?*’

If the model answered ‘no’ for an image without a stabilization system, it confirmed correct recognition.

If it answered ‘yes’ incorrectly, it indicated a misinterpretation of bone structures or artifacts as a stabilization system.

3.Identification of the Stabilization System Type

Question: ‘*If yes, what type of stabilization system is visible?*’

For images without a stabilization system, the correct response was ‘none’ or ‘not applicable’. If the model identified a system where none was present, it was considered a misclassification.

For images with stabilization, the model was expected to specify whether the system was growing (MCGR) or non-growing (PSF).


*Evaluation Criteria*


Each image was processed individually, and the model was asked the three questions in sequence. Only three key pieces of information were extracted from its responses:Presence of a stabilization system (if applicable),Type of stabilization system (growing vs. non-growing),Specific system identification (MCGR or PSF, if applicable).
*Scoring System*

For images without a stabilization system:Correctly identifying the absence of a system → 1 pointIncorrectly identifying a system → 0 pointsCorrectly stating no type/system was present → 1 pointIncorrectly specifying a type/system → 0 points

For images with a stabilization system:Correctly detecting the presence of a system → 1 pointFailing to detect a system → 0 pointsCorrectly identifying the type (growing/non-growing) → 1 pointIncorrectly classifying the type or failing to specify it → 0 pointsCorrectly recognizing the specific system (MCGR or PSF) → 1 pointIncorrectly identifying the system (e.g., misclassifying MCGR as VEPTR) or failing to specify → 0 points

To assess the model’s consistency and reliability, the entire study was repeated three times to evaluate whether the AI provided repeatable and stable responses.

The classification of scoliosis followed the AO Spine criteria, and the therapeutic approach was determined based on AO Spine qualifications for surgical intervention (Figure 2).

### 2.3. BiomedCLIP Methodology

#### 2.3.1. Model Selection and Adaptation

Pediatric spine X-ray images were classified across three conditions—pre-treatment, post-targeted stabilization system, and post-MAGEC system application—utilizing the Biomed CLIP model. Biomed CLIP, a refined variant of the Contrastive Language–Image Pretraining (CLIP) model, was selected due to its demonstrated superior performance in zero-shot classification of complex biomedical imagery, surpassing alternatives such as DALL-E, MedCLIP, and PubMedCLIP. Its open-source accessibility further facilitated its adoption for this research. To augment its specialization in pediatric spinal image analysis, Biomed CLIP underwent fine-tuning on the PMC-15 dataset. This comprehensive dataset comprises 15 million biomedical image−text pairs derived from 4.4 million scientific articles, encompassing a diverse array of medical specialties, thereby providing a robust foundation for model training.

#### 2.3.2. Hyperparameters

The fine-tuning of Biomed CLIP followed the same hyperparameters as those used for training the original CLIP ViT-B/16 model (Table 1). 

#### 2.3.3. Computational Infrastructure

Computational tasks were executed on the RunPod platform, employing two NVIDIA L40S GPUs, 16 vCPUs, 124 GB of RAM, and 96 GB VRAM. A standardized software environment was instantiated via a Docker image (pytorch:2.1.0-py3.10-cuda11.8.0-devel-ubuntu22.04), pre-configured with requisite CUDA and PyTorch modules to facilitate model training and subsequent evaluation.

#### 2.3.4. Model Architecture and Training Protocol

The Biomed CLIP architecture incorporates a Vision Transformer (ViT-B/16) as its vision encoder and PubMedBERT for text encoding. The ViT-B/16, selected for its superior efficiency relative to smaller variants, segments input images into 16 × 16 pixel patches, natively supporting a 224 × 224 pixel image size. While experimentation with a 384 × 384 pixel resolution indicated enhanced retrieval capabilities, this was accompanied by a commensurate increase in pre-training duration. The text encoder’s context window was extended to 256 tokens, from the conventional 77, to accommodate the verbose nature of biomedical image annotations. The ViT-B/16 vision encoder comprises 86 million trainable parameters. Model training was conducted on a cluster of 16 NVIDIA A100 or V100 GPUs, incorporating techniques such as gradient checkpointing and automatic mixed precision to optimize memory utilization and enhance computational throughput.

#### 2.3.5. Data and Performance Evaluation

Model performance was directly evaluated against a curated database of anonymized pediatric X-ray images. This approach enabled iterative assessment across various treatment stages and hardware configurations. The evaluation protocol prioritized the model’s classification accuracy and generalization capacity to unseen medical image data, thus providing a rigorous assessment of both the robustness and clinical applicability of Biomed CLIP.

The evaluation was conducted by interfacing directly with a database of anonymized X-ray images. This setup allowed for an iterative assessment of the model’s performance across different stages of treatment and varying hardware configurations. The evaluation focused on the model’s accuracy in image classification and its ability to generalize across unseen medical image data. This methodology not only tests the robustness of Biomed CLIP, but also its applicability to real-world medical diagnostics. For this task, descriptive text labels were prepared to represent the classification of each category:
First Category:
‘This is an image of spine with stabilization system’‘This is an image of spine without stabilization system’Second Category:
‘This is an image of spine with growing stabilization system’‘This is an image of spine with non-growing stabilization system’‘This is an image of spine without stabilization system’Third Category:
‘This is an image of spine with magnetic control growing rod system (MCGR)’‘This is an image of spine with posterior spinal fusion system (PSF)’‘This is an image of spine without stabilization system’

Each X-ray image was preprocessed through normalization before being input into the model. The model then computed the probability of the image belonging to each scoliosis category, with a confidence score ranging from 0 to 1. This score was used as a quantitative measure of the model’s certainty in its predictions.

The evaluation emphasized accuracy in image classification and the ability to generalize across unseen medical image data, testing both the robustness and real-world applicability of Biomed CLIP in clinical diagnostics.

### 2.4. Statistical Analysis

The detection rates of spinal stabilization systems were reported as count (n) and percentage (5) for binary classification tasks and as mean (M) and standard error (SE) for proportion-based analyses.

#### 2.4.1. Agreement Assessment

The assessment of inter-rater reliability for binary variables with multiple measurements was conducted using Gwet’s Agreement Coefficient 1 (AC1), due to its robustness in handling imbalanced category distributions and its reduced sensitivity to high agreement levels [22,23,24]. Gwet’s AC1 was then computed using Formula (1):(1)$$AC1=\frac{pa-pe}{1-pe}$$
where pa is the observed agreement estimated by (2):(2)$$pa=\frac{number\;of\;agreements}{total\;number\;of\;ratings}$$

The expected agreement by chance *pe* was conducted using (3):(3)$$pe=\sum_{k=1}^{C}{\left(\frac{1}{C}\right)}^{2}\;$$
where *C* is the number of categories.

Described by (1)–(3), AC1 remains robust and interpretable, not artificially deflated by the prevalence effect, thereby providing a more accurate reflection of inter-rater reliability, making it the optimal choice for the current analysis [25].

#### 2.4.2. Estimation Difference in Identification Rates Between AIs

To evaluate the significance of differences in detection proportions between GPT-4o and BiomedCLIP AI, a two-tailed z-test for the difference in proportions was conducted.

The standard error (SE) of the difference between the two means was computed using Formula (4):(4)$$SE=\sqrt{{SE}_{1}^{2}+{SE}_{2}^{2}},$$
where *SE*_1_ and *SE*_2_ represent the standard errors of the detection rates for GPT-4o and BiomedClip, respectively.

To determine the statistical significance of the difference, a z-score was calculated by (5):(5)$$z=\frac{M1-M2}{SE},$$
where *M*_1_ and *M*_2_ are the mean detection rates of GPT-4o and BiomedClip, respectively.

Finally, the *p*-value was derived from the standard normal distribution based on the calculated z-score. As a two-tailed test was used, the *p*-value was obtained according to (6):
*p* = 2 × *P*(*Z* > |*z*|),
(6)

where *P*(*Z* > |*z*|) represents the probability that a standard normal variable (*Z*) takes a value greater than the absolute value of the calculated z-score.

A significance threshold of α = 0.05 was applied to determine whether the observed differences were statistically significant.

#### 2.4.3. Statistical Tool

Analyses were conducted using the R Statistical language (version 4.3.3; [26]) on Windows 11 Pro 64 bit (build 22631), using the packages irrCAC (version 1.0; [27]) report (version 0.5.8; [28]), patchwork (version 1.2.0; [29]), gtsummary (version 1.7.2; [30]), ggplot2 (version 3.5.0; [31]), and dplyr (version 1.1.4; [32]).

## 3. Results

### 3.1. Comparative Sensitivity and Consistency in Detecting Spinal Stabilization Systems

The performance of GPT-4o and BiomedCLIP in detecting and classifying spinal stabilization systems (SSS) in 270 posturographic X-ray images (93 No Stabilization, 97 PSF, 80 MCGR) was evaluated using sensitivity (proportion of correct responses), Gwet’s AC1 for inter-rater agreement, and Cohen’s h for effect size (Table 2). A summary of findings aligned with the research hypotheses (H1: accuracy, H2: consistency, H3: classification) is presented in Table 3.

Detection of stabilization systems (H1: Accuracy)

GPT-4o demonstrated near-perfect sensitivity (97–100%) in detecting the presence or absence of SSS across all types (No Stabilization: 97.0%, 95% *CI* 94.0–100.0; PSF: 100.0%, 95% *CI* 100.0–100.0; MCGR: 100.0%, 95% *CI* 100.0–100.0), significantly outperforming BiomedCLIP (No Stabilization: 99.6%, 95% CI 99.6–99.6, *z* = 2.7, *p* = 0.007, *h* = −0.21; PSF: 5.1%, 95% *CI* 3.9–6.3, *z* = 95.0, *p* < 0.001, *h* = 2.05; MCGR: 6.6%, 95% *CI* 5.5–7.7, *z* = 93.0, *p* < 0.001, *h* = 1.95). The small difference in No Stabilization detection (2.6%, small effect) was clinically negligible, as both models excelled in this task. However, GPT-4o’s superior sensitivity for PSF and MCGR detection (large effect sizes) highlights its reliability for initial screening in clinical settings (Table 3).

Consistency of responses (H2)

BiomedCLIP exhibited perfect consistency (AC1 = 1.00) across all tasks, reflecting its deterministic classification approach. In contrast, GPT-4o showed a high but variable agreement, particularly in complex tasks. For No Stabilization and PSF, GPT-4o maintained strong agreement (AC1 = 0.97–1.00), but consistency dropped for MCGR Type (AC1 = 0.32, 95% *CI* 0.16–0.49) and Kind (AC1 = 0.50, 95% *CI* 0.36–0.64) classifications. Sensitivity differences showed a medium effect for No Stabilization Type (30.1%, *h* = 0.68), a large effect for PSF Type (97.4%, *h* = 2.41), and a small effect for MCGR Type (3.7%, *h* = 0.08), indicating variable clinical relevance (Table 3). This supports the conclusion that BiomedCLIP is more consistent, while GPT-4o’s probabilistic nature leads to response fluctuations.

Classification of SSS type and kind (H3)

For classifying SSS types (Growing vs. Non-growing), GPT-4o outperformed BiomedCLIP for PSF (98.0% vs. 0.6%, *z* = 37.5, *p* < 0.001, *h* = 2.41) but showed a comparable performance for MCGR (70.0% vs. 66.3%, z = 0.27, *p* = 0.790, h = 0.077). For specific system identification (Kind), GPT-4o was more accurate for PSF (84.0% vs. 11.0%, *z* = 36.5, *p* < 0.001, *h* = 1.18), but struggled with MCGR (18.0% vs. 4.2%, z = 1.40, *p* = 0.160, *h* = 0.433), reflecting challenges in distinguishing morphologically similar systems. Large effect sizes for PSF tasks confirmed GPT-4o’s clinical advantage, while smaller effects for MCGR tasks inferred that both models need optimization (Table 3). These findings indicate that GPT-4o is more effective for initial detection and broad classification, while BiomedCLIP’s structured approach offers potential for detailed classification with improved training.

### 3.2. Comparison of Mean Sensitivity Levels for GPT-4o’s Detection of Spinal Stabilization Systems

The proportion of proper responses in Table 1 is equivalent to sensitivity, reflecting the true positive rate of GPT-4o’s detection of spinal stabilization systems (SSS) in advanced scoliosis (≥40°) using posturographic radiographic images (PRIs). Figure 3 compares the mean sensitivities across the type of PRIs of the SSS and the type of request about SSS. The results are stratified by the type of SSS (Absence, PSF, MGCR) and the type of request (Presence, Type, Kind), with error bars indicating the standard error (SE) of the mean sensitivity. This analysis demonstrates the performance and variability of GPT-4o’s detection capabilities, providing a comprehensive evaluation of its accuracy and reliability in identifying SSS under varying conditions.

### 3.3. Analysis of the Performance and Interrater Agreement of BiomedClip AI in Identifying Spinal Stabilization Systems in Severe Scoliosis (≥40°) Using Posturographic Radiographic Imaging

In contrast to GPT-4o, BiomedCLIP AI enabled the estimation of the identification proportion for the queried options with a granularity of 0.01, with the total sum equaling 1.00. This approach allowed for greater precision in performance assessment.

A key distinction was the consistency of proportion estimates across repeated analyses of the same images, resulting in perfect agreement (AC1 = 1) in the identification of all types of stabilization systems.

#### 3.3.1. Analysis of the Performance in Identifying Posturographic Radiographic Images with No Stabilization Systems

BiomedCLIP AI’s performance differed notably depending on whether the task was simply to detect the presence or absence of stabilization hardware, to identify the ‘type’ of stabilization, or to specify the ‘kind’ of construct.

Under ‘Presence of stabilization’ requests, the model achieved a mean correct classification rate of 0.996 (SE = 0.00) for images containing no stabilization. In other words, the AI system successfully ruled out stabilization in nearly 99.6% of cases, with only 0.4% of images incorrectly flagged as having some form of stabilization systems (see Figure 4 for visualization).

When tasked with specifying the ‘type’ of stabilization (Growing rod, No growing rod, or No stabilization), BiomedCLIP AI continued to classify the majority of true No-stabilization images correctly (0.669, SE = 0.02). However, 0.106 of these images were mistakenly labeled as ‘Growing rod’, and 0.225 were labeled as No growing rod, indicating that the model still confused some images that should have been confidently identified as ‘No stabilization’.

Performance diminishes further under ‘kind’ classifications (PSF, MCGR, No stabilization), where BiomedCLIP AI correctly retained the no-stabilization label in 0.302 (SE = 0.02) of cases, but mislabeled 0.673 as MCGR and 0.0252 as PSF. This observation highlights that while the BiomedCLIP AI is highly reliable for straightforward detection of absence or presence, it becomes less precise when finer distinctions are required, particularly if the request demands a specific ‘kind’ of hardware category.

#### 3.3.2. Analysis of the Performance in Identifying Posturographic Radiographic Images with PSF Stabilization System

BiomedClip AI’s ability to detect a posterior spinal fusion (PSF) in severe scoliosis (≥40°) posturographic imaging remained suboptimal when dissected by specific classification requests. Under the most basic ‘Detection’ protocol, the system correctly identified ‘Stabilization’ in only 5.10% (SE = 0.00568) of cases, while it incorrectly assigned ‘No stabilization’ 94.90% of the time (see Figure 5 for visualization results).

In the subsequent ‘Type’ categorization, correctness hinged on labeling these PSF images as ‘No growing rod’; the model achieved a mere 0.64% success rate (SE = 0.00128), with the remaining classifications scattered among ‘Growing rod’ or ‘No stabilization’. At the most granular ‘Kind’ level—where a precise PSF designation was required—the model attained a correctness rate of only 11.0% (SE = 0.0183), while disproportionately mislabeling many of these images as ‘No stabilization’ (86.2%) or ‘MCGR’ (2.77%).

These findings underscore BiomedClip AI’s marked tendency to overlook PSF constructs and misinterpret them, especially in more specialized requests that demand accurate differentiation between various fixation categories. Although the system excelled in other contexts (e.g., ruling out hardware for individuals genuinely lacking instrumentation), it struggled here to recognize key radiographic hallmarks of a posterior spinal fusion.

From a clinical perspective, the low detection and classification rates for a procedure as prevalent and standardized as PSF may prompt the need for further refinement of BiomedClip’s training data, specifically incorporating robust examples of routine posterior fusion hardware in advanced scoliosis.

#### 3.3.3. Analysis of the Performance in Identifying Posturographic Radiographic Images with MCGR Stabilization Systems

BiomedClip AI displayed a pronounced discrepancy in classifying MCGR-based stabilization. When merely tasked with detecting whether spinal hardware was present, the model labelled only 6.62% (SE = 0.00564) of MCGR images correctly as ‘Stabilization’ and misclassified the remaining 93.38% as ‘No stabilization’ (see Figure 6 for visualization results).

Once directed to identify the ‘type’ of hardware, it correctly assigned ‘Growing rod’ status to 66.3% (SE = 0.0177) of these same images, suggesting that once it (occasionally) recognized the presence of an implant, the system was comparatively adept at recognizing it as a ‘growing’ construct.

However, under the most granular ‘kind’ classification, only 4.16% (SE = 0.0150) were labeled accurately as ‘MCGR’, while 86.8% remained misclassified as ‘No stabilization’, with another 9.05% classified as ‘PSF’. The net effect ws that although BiomedClip AI could sometimes characterize the hardware as ‘growing’ once it surmised stabilization waas present, its overall performance for consistently detecting MCGR constructs remained suboptimal.

Clinically, these results indicate that, in the context of advanced scoliosis (≥40°), the model’s failure to recognize the initial presence of hardware compromised the reliability of subsequent designations. Such gaps highlight the need for further refinement of the system’s training data—especially involving subtle morphological traits of MCGR rods—to ensure both accurate detection and correct categorization of complex dynamic instrumentation.

### 3.4. Comparative Sensitivity Analysis of BiomedClip AI and GPT-4o in Spinal Instrumentation Classification

#### 3.4.1. Posturographic Radiographic Images with No Stabilization Systems

BiomedClip AI demonstrated significantly higher sensitivity in the presence-versus-absence detection of spinal instrumentation (M = 99.7%, SE = 0) compared to GPT-4o (M = 97%, SE = 1%), with a statistically significant difference (z = 2.7, *p* = 0.007).

However, as classification complexity increased, GPT-4o maintained a consistently high sensitivity (97%), whereas BiomedClip AI’s performance declined substantially. For the ‘type’ classification task, BiomedClip AI achieved M = 66.9%, SE = 2%, significantly lower than GPT-4o (M = 97%, SE = 1%, z = −13.44, *p* < 0.001).

This performance disparity was even more pronounced in the ‘kind’ classification task, where BiomedClip AI’s sensitivity dropped to M = 30.2%, SE = 3.2%, while GPT-4o remained stable at M = 97%, SE = 1% (z = −19.94, *p* < 0.001).

These findings indicate that while BiomedClip AI excelled in binary classification tasks, GPT-4o provided superior reliability and stability in more complex hierarchical classifications. This suggests that for clinical applications requiring detailed differentiation of spinal hardware, GPT-4o may offer a more robust and consistent solution.

#### 3.4.2. Posturographic Radiographic Images with PSF Stabilization Systems

GPT-4o demonstrated a significantly superior performance compared to BiomedClip AI in detecting spinal stabilization systems across multiple classifications. In identifying the presence of stabilization systems, GPT-4o achieved perfect sensitivity (M = 1.00, SE = 0.00), markedly outperforming BiomedClip (M = 0.05, SE = 0.01), with a highly statistically significant difference (z = 95.00, *p* < 0.001).

Similarly, for the detection of non-growing rod systems, GPT-4o exhibited near-perfect sensitivity (M = 0.98, SE = 0.00), significantly surpassing BiomedClip (M = 0.23, SE = 0.02), as reflected in a highly significant statistical difference (z = 37.5, *p* < 0.001). A comparable trend was observed in the classification of PSF systems, where GPT-4o (M = 0.84, SE = 0.01) outperformed BiomedClip (M = 0.11, SE = 0.02), with a highly significant difference (z = 36.5, *p* < 0.001).

The above findings stress the substantial advantage of GPT-4o in accurately detecting and classifying spinal stabilization systems, highlighting its potential as a more effective tool for clinical decision-making, particularly in the management of complex scoliosis cases compared to BiomedClip.

#### 3.4.3. Posturographic Radiographic Images with MCGR Stabilization Systems

GPT-4o demonstrated significantly superior accuracy in detecting stabilization systems, achieving perfect sensitivity (M = 1.00, SE = 0.00) compared to BiomedClip (M = 0.07, SE = 0.01), with a highly statistically significant difference (z = 93, *p* < 0.001). However, in the detection of growing-rod systems, GPT-4o (M = 0.70, SE = 0.15) and BiomedClip (M = 0.66, SE = 0.02) exhibited a comparable performance, with no statistically significant difference (z = 0.27, *p* = 0.790). For MCGR systems, GPT-4o (M = 0.18, SE = 0.10) outperformed BiomedClip (M = 0.04, SE = 0.02), although the difference did not reach statistical significance (z = 1.40, *p* = 0.160), suggesting a trend toward an improved performance.

These findings highlight GPT-4o’s superior capability in detecting stabilization systems, reinforcing its potential as a more reliable tool for identifying such hardware in clinical practice. In contrast, for rowing rod systems, both models performed similarly, indicating that either may be effectively utilized in this context. However, the overall performance for MCGR classification remained suboptimal, suggesting that both models struggled with accurately identifying these systems.

## 4. Discussion

The findings of this study provide partial confirmation of our initial hypotheses, revealing both strengths and limitations in the performance of ChatGPT-4o and BiomedCLIP in detecting and classifying spinal stabilization systems in posturographic X-ray images.

Regarding H1 (accuracy in detecting stabilization systems), the results contradict our initial assumption, as ChatGPT-4o consistently outperformed BiomedCLIP in detecting the presence of spinal hardware, achieving near-perfect sensitivity (97–100%), while BiomedCLIP misclassified stabilization in a substantial number of cases (e.g., PSF detection: 5.10%, MCGR detection: 6.62). These results indicate that ChatGPT-4o, although not specifically trained for medical imaging, performs more reliably when identifying the presence or absence of spinal hardware.

Conversely, H2 (consistency of model responses) was confirmed, as BiomedCLIP exhibited perfect agreement (AC1 = 1.00) across repeated classifications, while ChatGPT-4o displayed variability in its responses, particularly in more complex classification tasks, such as distinguishing between stabilization types (Growing vs. Non-Growing) and specific systems (MCGR vs. PSF). This variability suggests that while ChatGPT-4o is highly effective in initial detection, its decision-making process lacks stability across repeated analyses.

The results for H3 (effectiveness in classifying the type and nature of the stabilization system) were mixed. ChatGPT-4o effectively identified the presence of stabilization systems but struggled with the precise classification of type and system (e.g., MCGR classification accuracy of only 18%). BiomedCLIP, despite its limitations in basic detection, performed slightly better in distinguishing specific types of stabilization systems once hardware was recognized, albeit with suboptimal accuracy. This indicates that while ChatGPT-4o is more effective at recognizing the presence of stabilization, BiomedCLIP provides more structured classification but requires further optimization to improve accuracy.

### 4.1. Generalist vs. Specialist AI: Understanding the Superior Detection Performance of ChatGPT-4o in Spinal Stabilization System Identification

The results of our study indicate that ChatGPT-4o outperformed BiomedCLIP in detecting the presence of spinal stabilization systems, contradicting H1, which hypothesized that BiomedCLIP would demonstrate superior accuracy due to its specialization in medical image analysis. Several factors may explain this unexpected finding, aligning with recent insights from AI research.

The finding that ChatGPT-4o outperformed BiomedCLIP in detecting spinal stabilization systems—despite contradicting Hypothesis 1—invites further interpretation. One possible explanation is the trade-off between generalization and specialization—ChatGPT-4o, as a generalist model, was trained on a vast and diverse dataset encompassing both text and images, which may have endowed it with the ability to detect features beyond the medical context. Its ability to detect broad patterns and high-contrast features, like stabilization systems, may explain its strong performance in binary classification tasks. In contrast, BiomedCLIP was fine-tuned for specific tasks related to matching medical images to text, suggesting a greater reliance on textual descriptions in the classification process. This study tested the model solely on image data, which could limit its ability to precisely recognize stabilization systems. A similar phenomenon has been described in the literature, indicating that generalist models like ChatGPT-4o can achieve competitive results in X-ray image analysis, particularly in tasks involving the detection of distinct contrast structures, without the need for deep semantic interpretation of the image. In summary, BiomedCLIP utilized stringent decision thresholds, while ChatGPT-4o likely had exposure to more diverse radiographic images, including variable contrast, noise levels, and different anatomical conditions, which may have enhanced its adaptation to unknown clinical cases [32]. Conversely, despite being tuned to the PMC-15 dataset (15 million image-text pairs), BiomedCLIP may not have been sufficiently trained on posturographic X-ray images of the spine with stabilization implants. This was confirmed by analyses indicating that the diversity of training datasets is crucial for models’ generalization capabilities. Significant roles may also be played by differences in feature extraction mechanisms—ChatGPT-4o likely utilizes a vision model based on the transformer architecture, similar to CLIP or GPT-Vision, known for efficiently detecting objects even in complex scenes [32]. It is possible that its classification relies more heavily on detecting features of shape, contrast, and texture, facilitating the identification of metallic structures such as MCGR or PSF systems, regardless of their orientation or intensity levels. In contrast, BiomedCLIP, despite using a ViT-based vision encoder, may be optimized primarily for multimodal search, which limits its effectiveness in purely visual tasks. The results indicate that ViT models tuned to text−image retrieval may not achieve a maximum performance in tasks requiring independent image analysis [33]. Finally, key aspects may include differences in models’ confidence thresholds and decision boundaries. ChatGPT-4o might have employed more lenient classification thresholds, resulting in high sensitivity in detecting stabilization systems, even with some level of uncertainty. This approach could increase the number of correctly identified cases, although, in other contexts, it might lead to a higher number of false positive results. In contrast, BiomedCLIP may have used more rigorous decision boundaries, which increased its precision, but might also have led to a higher number of missed detections (false negatives). The work *Generative Artificial Intelligence in Anatomic Pathology* suggests that more conservative models, operating at high decision thresholds, may more effectively avoid false positive errors, but might also be more susceptible to missing significant detections [34]. Jegham et al. suggested that ChatGPT-4o employs a well-calibrated mechanism for recognizing uncertainty, which could explain its high efficacy in detection tasks. Overall, the results of this study indicate that generalist models like ChatGPT-4o can achieve high effectiveness in medical image classification due to their broad exposure to diverse data and more flexible feature extraction mechanisms [35]. However, specialist models, although often characterized by greater precision in their niche applications, might be more limited in tasks requiring independent visual analysis without textual support.

### 4.2. Possible Explanations for the Confirmation of H2 (Model Response Consistency)

The confirmation of H2, which predicted that BiomedCLIP would demonstrate greater consistency in classifying spinal stabilization systems than ChatGPT-4o, can be attributed to several key differences in their architectures, training methodologies, and response mechanisms. Below are potential explanations for why BiomedCLIP exhibited perfect agreement (AC1 = 1.00) across repeated classifications, while ChatGPT-4o showed response variability, particularly in complex classification tasks.

### 4.3. Deterministic vs. Probabilistic Decision-Making

Research on multimodal AI models like BiomedCLIP and ChatGPT-4o has highlighted fundamental differences in their approach to image classification, stemming from distinct decision-making mechanisms. BiomedCLIP, optimized for medical imaging analysis, exhibits deterministic behavior—its classifications are repeatable and consistent, regardless of the number of iterations with the same image. This predictive structure is characteristic of specialist models trained on structured datasets, where precise and consistent image interpretation is critical [36]. In contrast, language−visual models like ChatGPT-4o introduce a probabilistic component to image analysis. As a transformer-based LLM, GPT-4o does not operate like a traditional deterministic classifier—its predictions can vary depending on temperature settings, token sampling methods, and subtle differences in data preprocessing [37]. This mechanism means that while ChatGPT-4o may show a high sensitivity in initial structural detection, its specific classifications may fluctuate between subsequent analyses of the same photo. Probabilistic models are commonly used in radiology and diagnostic description generation, as their ability to interpret data stems from estimating probability distributions between possible outcomes and actual labels [38]. Moreover, vision−language analysis indicates that the classification mechanisms in models like GPT-4o are based on comparing probability distributions, reinforcing the thesis about their probabilistic nature [39]. In the context of medical applications, studies on transformer-based models suggest that although they may exhibit some level of determinism in task planning, their final responses to diagnostic questions remain variable depending on the input data and result generation methods [40]. These findings highlight key differences between deterministic and probabilistic models: BiomedCLIP delivers consistent results, while ChatGPT-4o offers flexibility, but may vary depending on input conditions.

### 4.4. Differences in Model Training Objectives and Architecture

Studies on multimodal AI models like BiomedCLIP and GPT-4o reveal fundamental differences in their training objectives, architectures, and image processing methods. BiomedCLIP, designed for precise image−text alignment, relies on contrastive learning and uses fixed embeddings and a highly structured visual feature extraction pipeline, making its classifications consistent and repeatable upon multiple analyses of the same image [41]. Its deterministic nature has been confirmed in studies on the stability of specialist models in image analysis, which indicate that BiomedCLIP maintains a high precision by tightly matching image features to its prior representations [42]. In contrast, ChatGPT-4o, as a general-purpose model, does not rely on static embeddings in the same way. Its visual system interprets images dynamically, in the context of prior experiences and a broad range of training data, which can lead to slight variations in the classification of the same image across different iterations [43]. The literature emphasizes that such models may struggle with the fine classification of hierarchical structures, like distinguishing MCGR and PSF systems in X-ray images, due to their more abstract feature extraction strategy [44]. Unlike BiomedCLIP, tuned for medical image analysis, it exhibits higher stability in text−visual image classification, but its effectiveness may be limited in tasks not requiring direct linkage to textual data [45]. In summary, these studies confirm that BiomedCLIP, as a specialist model, achieves high precision by tightly matching visual features, while GPT-4o, with its generative nature, applies a dynamic approach to image analysis, which may result in some fluctuations in the classification of the same image between iterations.

### 4.5. Handling of Complex Morphological Features in X-Ray Images

Research on multimodal AI models indicates that BiomedCLIP offers greater consistency in radiographic classification than LLM-based models like ChatGPT-4o, due to its structured approach to feature extraction. As a model optimized for medical image analysis, BiomedCLIP uses precise feature mapping and is fine-tuned to specialized datasets such as PMC-15 (15 million image−text pairs), which minimizes classification ambiguity and allows for more accurate assignment of images to specific diagnostic categories [46]. Although this approach permits a degree of flexibility, it means that medical image classification may be more prone to errors, especially in cases of structures with high density and morphological similarity [47]. Furthermore, the literature suggests that in LLM models, image classification often relies on image−text matching strategies, which can lead to ambiguous results in strictly visual tasks where textual context is not available [48]. This phenomenon is particularly noticeable in the analysis of X-ray images with high-density anatomical structures, where LLM-based models exhibit greater variability in results compared to BiomedCLIP, which uses dedicated medical encoders, such as BiomedCLIP-CXR, to reduce classification discrepancies [49]. Additionally, the heuristics used by LLM models, including GPT-4o, are not precise enough to achieve effectiveness comparable to deep learning systems designed specifically for medical image analysis [50]. In the context of clinical applications, this suggests that models like BiomedCLIP may be more suitable for precise diagnostic tasks, while generative models, although potentially useful in exploratory tasks, may require additional mechanisms to stabilize results in order to obtain repeatable and reliable predictions.

### 4.6. Influence of Decision Thresholds and Ambiguity Tolerance

The analysis of decision thresholds used by BiomedCLIP and ChatGPT-4o points to significant differences in their approach to medical image classification, particularly regarding sensitivity in detection and prediction stability. BiomedCLIP appears to employ a stringent certainty threshold for classification, meaning it assigns labels only when the forecast meets a high criterion of reliability. This approach increases result stability and reduces the number of incorrect classifications, but it may also lead to reduced sensitivity in detecting less distinct radiographic structures [51]. Since BiomedCLIP has been fine-tuned to specialized medical datasets, its classification decisions are strongly dependent on previously learned patterns, which may limit its ability to adapt to new clinical cases and rare anomalies [52]. Conversely, ChatGPT-4o features a more flexible certainty threshold, which allows it to more effectively detect the presence of spinal stabilization systems, but also means its classifications may be less stable. As a generative model based on a probabilistic approach to image processing, GPT-4o tends to exhibit greater sensitivity in classification tasks, increasing the number of correct detections, but also making it more susceptible to erroneous identifications in cases of structurally similar objects [53]. This is because LLM models like GPT-4o apply a dynamic approach to classification decisions, where prediction is based on probability estimation and gradual alignment with the most likely category, which can lead to greater variability in results compared to deterministic models like BiomedCLIP [43]. Research on multimodal AI systems indicates that generalist language−visual models can achieve high effectiveness in detecting broad object categories, but lose precision when classification requires distinguishing fine differences between structures [38]. These findings underscore the critical role of selecting a certainty threshold in designing AI models for medical applications, and suggest that the effectiveness of image analysis systems can be optimized by combining the advantages of both specialist and generalist models.

### 4.7. Explaining the Results for H3: Effectiveness in Classifying the Type and Nature of Stabilization Systems

The findings for H3 suggest that ChatGPT-4o was highly effective at detecting the presence of stabilization systems, but struggled with detailed classification, whereas BiomedCLIP, despite lower overall detection rates, performed slightly better in differentiating between stabilization system types (Growing vs. Non-Growing) and specific systems (MCGR vs. PSF), albeit with suboptimal accuracy. Several factors likely contributed to these results.

### 4.8. BiomedCLIP’s Dependence on Structured Feature Representations

BiomedCLIP features a structured approach to classification, enabling it to more accurately determine the type and structure of a spinal stabilization system once detected. Unlike general-purpose models such as ChatGPT-4o, BiomedCLIP employs a more logical and hierarchical classification scheme, which allows it to achieve a higher accuracy when differentiating between MCGR and PSF systems [42]. BiomedCLIP’s lower implant detection accuracy likely results from its primary design focus on image-to-text matching, rather than direct object recognition. Research on multimodal AI systems shows that models integrating textual analysis perform better in structural classification and contextually demanding tasks, but struggle with object classification when the textual input component is absent [54]. In the context of medical classification, this may mean that BiomedCLIP is better suited for tasks where image analysis is supported by descriptive metadata, allowing it to more finely distinguish similar anatomical structures. In contrast, ChatGPT-4o relies mainly on visual patterns and more flexible heuristic classification, which enables it to effectively detect high-contrast objects, but may lead to greater variability in their subsequent classification. These differences highlight that while BiomedCLIP achieves greater accuracy in fine-grained classification after initial object detection, generative models like ChatGPT-4o may perform better in tasks requiring high-sensitivity detection, but with a greater risk of incorrect detailed classifications. This suggests that an optimal approach to medical image analysis might involve a hybrid application of both types of models—using a general model for structure detection and a specialized model like BiomedCLIP for precise hierarchical classification.

### 4.9. Difficulty in Differentiating Morphologically Similar Hardware

Spinal stabilization systems such as MCGR and PSF often exhibit significant visual convergence in posturographic X-ray images, especially when viewed from different angles or under varying contrast conditions. This structural similarity can pose a challenge for AI models like ChatGPT-4o, which largely base their classification decisions on shape heuristics rather than detailed morphological analysis. Research on radiological image interpretation shows that ChatGPT-4o often uses global visual features for decision-making, which can lead to erroneous classifications, especially when the analyzed objects exhibit significant visual similarity [55]. Similar limitations have been observed in other areas of medical diagnostics—for example, in zero-shot classification of lung diseases, where AI models struggled to identify unknown patterns without access to highly specialized datasets [56]. In contrast, multimodal models like BiomedCLIP may partially compensate for this issue by utilizing textual associations, which allows them to more precisely differentiate structures with a high degree of visual similarity. BiomedCLIP, fine-tuned based on large image−text pair datasets, likely uses contextual language connections to improve the hierarchical classification of stabilization systems, although its effectiveness remains limited due to inadequate coverage of these specific cases in training sets [57]. The literature has indicated that for diagnosing diseases with similar morphological progression, such as oral lichen planus, AI models, including ChatGPT-4o, struggled to differentiate subtle features of the disease without specialized training [58]. This points to a broader problem of generalist models, which despite high effectiveness in classifying structures with distinct contrast, may not perform equally well in tasks requiring detailed differentiation of similar objects.

### 4.10. ChatGPT-4o’s Variability in Classification of Specific Systems

The inconsistency in classification observed with ChatGPT-4o stems from its probabilistic nature, which causes its results to vary between iterations, even if the initial detection of the stabilization system was correct. This issue is particularly evident in hierarchical classification tasks, where the model must first determine the general category of an object and then precisely assign it to a specific subtype. In the classification of spinal stabilization systems MCGR and PSF, it was noted that although ChatGPT-4o could effectively detect the presence of an implant, its decisions regarding the specific type of system were less consistent and showed significant variability across iterations [59]. This phenomenon has been widely documented in studies on transformer-based models, which often exhibit classification instability in tasks requiring the recognition of subtle structural differences [34].

In anatomical classification, where distinguishing minor morphological differences is necessary, AI based on probabilistic mechanisms often changes its predictions depending on subtle changes in input data, which makes their effectiveness in classifying detailed subtypes lower than in general categories. A similar problem was observed in studies on taxonomic classification, where generative models, such as TaxonGPT, struggled with transitioning from broad categories to more detailed subdivisions, leading to reduced accuracy in subtype classification [60].

In the context of medical image analysis, this means that while ChatGPT-4o achieves high effectiveness in recognizing high-contrast structures, its classification at a more detailed level is less predictable. An additional factor affecting result variability is how transformer-based models interpret visual features depending on input conditions and sampling strategies. Research shows that such AI may be prone to inaccuracies, especially if a given image lacks clear features enabling precise assignment to a specific subcategory [61].

In the classification of spinal stabilization systems, this means that while ChatGPT-4o may correctly detect an implant, its classification of the specific system type is more susceptible to fluctuations arising from contextual differences in image processing. This phenomenon suggests that models with a probabilistic architecture may require additional mechanisms to stabilize outcomes in tasks requiring consistent hierarchical classification, particularly in the context of medical diagnostics, where precision and repeatability of results are critical.

### 4.11. Clinical Implications of GPT-4o’s Diagnostic Stability

The high stability of GPT-4o in detecting the presence of spinal stabilization systems across repeated analyses suggests important clinical implications. In particular, its consistent performance in binary classification tasks may support real-time decision-making in radiological triage, especially in resource-limited settings. Its ability to reliably identify the presence of metallic hardware could enhance workflows by pre-screening images for further specialist evaluation. While GPT-4o shows limitations in precise classification of hardware type, its robustness in identifying hardware presence highlights its potential as an effective support tool in early diagnostic pipelines or as a cross-verification step in clinical practice.

### 4.12. Analysis of BiomedCLIP Performance on Stabilization Classification

Initial evaluation of the BiomedCLIP model’s performance indicates a moderate capability in differentiating between the presence and absence of spinal stabilization interventions. The model demonstrates reasonable accuracy when classifying images as either exhibiting evidence of stabilization or showing no such intervention. However, a significant performance degradation is observed when the task requires finer-grained classification, specifically distinguishing between different types of stabilization systems.

In scenarios requiring differentiation among specific stabilization methods, the model exhibits a pronounced bias towards selecting labels containing the term ‘stabilization’, irrespective of other relevant label components. This behavior suggests a potential issue related to the semantic weighting of the term ‘stabilization’ within the model’s learned representations. The frequent occurrence of ‘stabilization’ in the training data, particularly in the context of the MCGR system (where the term is often used even when referring to alternative approaches), may have led to an overemphasis on this specific term. This creates an ‘archetype effect’, where the strong semantic association of ‘stabilization’ overshadows other, more nuanced, aspects of the image and label.

Furthermore, the use of negative labels, such as ‘no stabilization’, presents an additional challenge. Negation, while readily understood by humans, remains a significant difficulty for many machine learning models, particularly in natural language processing. One of the primary challenges arises from the complexity of negation scope detection, which affects tasks like sentiment analysis and text classification. Despite advancements in deep learning, accurately replicating human-like interpretations of negation remains an open problem [62]. The model may struggle to correctly interpret the negation of a concept (‘stabilization’) that has a strong, positive association in its training data. The inherent illogical nature—from the model’s perspective—of negating a strongly established concept likely contributes to the observed misclassifications.

It is important to emphasize that BiomedCLIP was not specifically trained on radiographic datasets featuring spinal stabilization hardware. As an open-source model developed for broad biomedical image–text alignment, its training corpus did not explicitly include posturographic X-ray images with metallic implants such as MCGR or PSF systems. Therefore, this study aimed not to critique a misapplication of the model, but rather to explore its potential when applied to a complex diagnostic task outside its original design scope. The observed limitations in implant detection may stem from the model’s lack of exposure to high-density artifacts commonly associated with surgical instrumentation, highlighting a broader challenge in transferring specialist AI models to new clinical domains without targeted fine-tuning. These findings underscore the necessity of dedicated retraining on implant-specific datasets if BiomedCLIP is to be effectively adapted for orthopedic diagnostics.

### 4.13. Study Limitations

This study has several limitations that should be considered when interpreting the results. First, although the dataset consisted of 270 posturographic X-ray images selected according to strict inclusion and exclusion criteria, the sample size remained relatively small compared to large-scale AI evaluation studies. The limited dataset may affect the generalizability of the findings, particularly regarding the AI models’ ability to handle a broader range of anatomical variations, image quality discrepancies, and cases with atypical spinal deformities.

Second, while the study included a rigorous three-stage evaluation process for AI-based image interpretation, the methodology inherently relies on textual responses generated by ChatGPT-4o, which are subject to variability due to the model’s probabilistic nature. This variability may not have been fully accounted for, even with repeated testing in this study. Future research should explore methods to quantify and mitigate variability in AI-generated responses to improve reproducibility.

Third, BiomedCLIP, although fine-tuned on the PMC-15 dataset, was not explicitly trained on pediatric posturographic spinal images, which could have influenced its performance in this domain. The model’s reliance on text−image alignment suggests that its effectiveness may be enhanced with additional fine-tuning on a dataset specifically curated for pediatric scoliosis imaging. Additionally, the model’s classification accuracy may have been impacted by the resolution and format of the X-ray images (JPEG, 2663 × 1277 px), as medical imaging AI models often benefit from higher-resolution DICOM formats that preserve more radiographic details.

Finally, the computational environment used for BiomedCLIP’s training and evaluation, while standardized, may differ from real-world clinical deployment conditions. Factors such as processing power, inference time, and real-time interpretability were not extensively analyzed in this study. Future work should assess how these AI models perform in clinical workflows, considering integration challenges, latency, and model interpretability for radiologists and orthopedic specialists.

Another important consideration is the broader limitation of using open-source AI models in clinical settings. Although models like GPT-4o and BiomedCLIP are accessible and promote innovation, they lack regulatory approval, clinical validation, and task-specific fine-tuning. These factors may limit their reliability and trustworthiness in high-stakes diagnostic contexts. Moreover, open-source models often rely on general datasets that do not adequately represent specialized medical scenarios, such as pediatric spinal stabilization. This study contributes to addressing these gaps by highlighting specific strengths and weaknesses of such models in a controlled evaluation, offering a foundation for future refinement and clinical adaptation.

Despite these limitations, the study provides valuable insights into the comparative performance of ChatGPT-4o and BiomedCLIP in spinal stabilization system detection and classification. Future research should expand dataset diversity, explore additional AI fine-tuning strategies, and investigate clinical feasibility to enhance the practical application of AI-driven scoliosis assessment.

### 4.14. Future Directions for Model Validation and Improvement

To ensure clinical relevance and broader applicability, further validation of AI models for spinal stabilization system detection is essential. Future research should focus on fine-tuning models like BiomedCLIP using larger, multicenter datasets specifically curated for pediatric spinal imaging with surgical implants. Additionally, combining generalist and specialist models may enhance diagnostic accuracy. Prospective clinical testing and integration into radiology workflows will be key to evaluating the real-world utility and safety of such systems.

## 5. Conclusions

This study provides a comparative evaluation of ChatGPT-4o and BiomedCLIP in the detection and classification of spinal stabilization systems on pediatric posturographic X-ray images. The findings highlight key differences between these AI models in terms of detection accuracy, classification consistency, and interpretability in a clinical context.

ChatGPT-4o demonstrated a higher sensitivity in detecting the presence of a spinal stabilization system compared to BiomedCLIP. However, its probabilistic nature resulted in classification inconsistencies, particularly in distinguishing between MCGR and PSF stabilization systems. The model’s reliance on general shape-based heuristics likely contributed to its misclassification of visually similar structures, reinforcing prior findings that generalist AI models may struggle with fine-grained medical classification tasks without extensive domain-specific training. Despite its high detection rate, ChatGPT-4o’s variable responses across repeated trials indicate a need for stability-enhancing mechanisms in clinical applications.

In contrast, BiomedCLIP exhibited greater classification consistency, particularly in hierarchical differentiation between stabilization system types. Its structured classification approach, combined with its image−text alignment capability, allowed for more precise subtype identification once a system was detected. However, its lower sensitivity in initial detection suggests that its performance was limited by dataset constraints and its primary optimization for image−text retrieval rather than pure visual analysis. While BiomedCLIP demonstrated advantages in structured classification, its dependency on specialized training data underscores the importance of domain-specific fine-tuning for optimal medical AI performance.

The study also underscores broader challenges associated with AI-based radiological interpretation, including model variability, dataset limitations, and the need for standardized evaluation protocols. The differences observed between ChatGPT-4o and BiomedCLIP suggest that neither model, in its current state, offers a fully reliable standalone solution for automated scoliosis assessment. Instead, a hybrid approach—leveraging the detection sensitivity of generalist models like ChatGPT-4o and the classification consistency of specialized models like BiomedCLIP—may offer a more robust framework for AI-assisted spinal imaging analysis.

Future research should focus on expanding dataset diversity, refining AI training methodologies, and integrating multimodal AI models into clinical workflows to improve both detection accuracy and classification reliability. Additionally, exploring methods to enhance model interpretability and reduce response variability will be crucial for increasing the clinical trustworthiness of AI-assisted radiological assessments. Despite its limitations, this study provides valuable insights into the capabilities and challenges of AI models in pediatric spinal imaging, paving the way for further advancements in automated scoliosis evaluation.

## Figures and Tables

**Figure 1 jcm-14-03282-f001:**
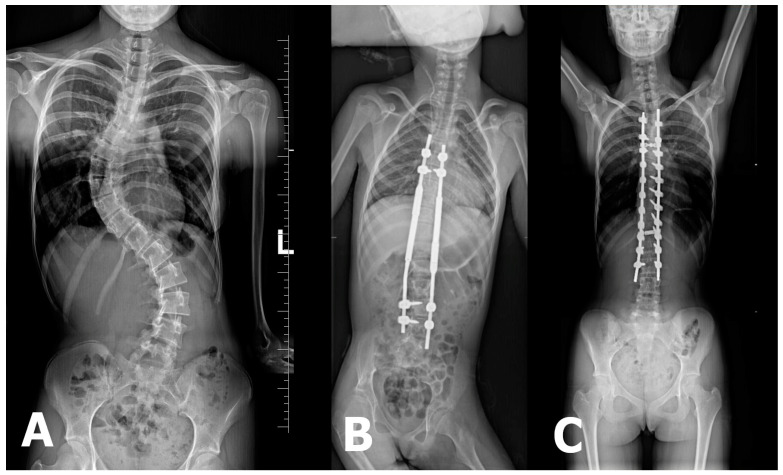
(**A**) X-ray image depicting severe double-curve scoliosis with Cobb angle measurements of 62° between L4/L5 and Th11/Th12 and 77° between Th11/Th12 and Th6/Th5. (**B**) X-ray image showing the magnetically controlled growing-rod (MCGR) system used for spinal stabilization. (**C**) X-ray image illustrating the final spinal stabilization procedure.

**Figure 2 jcm-14-03282-f002:**
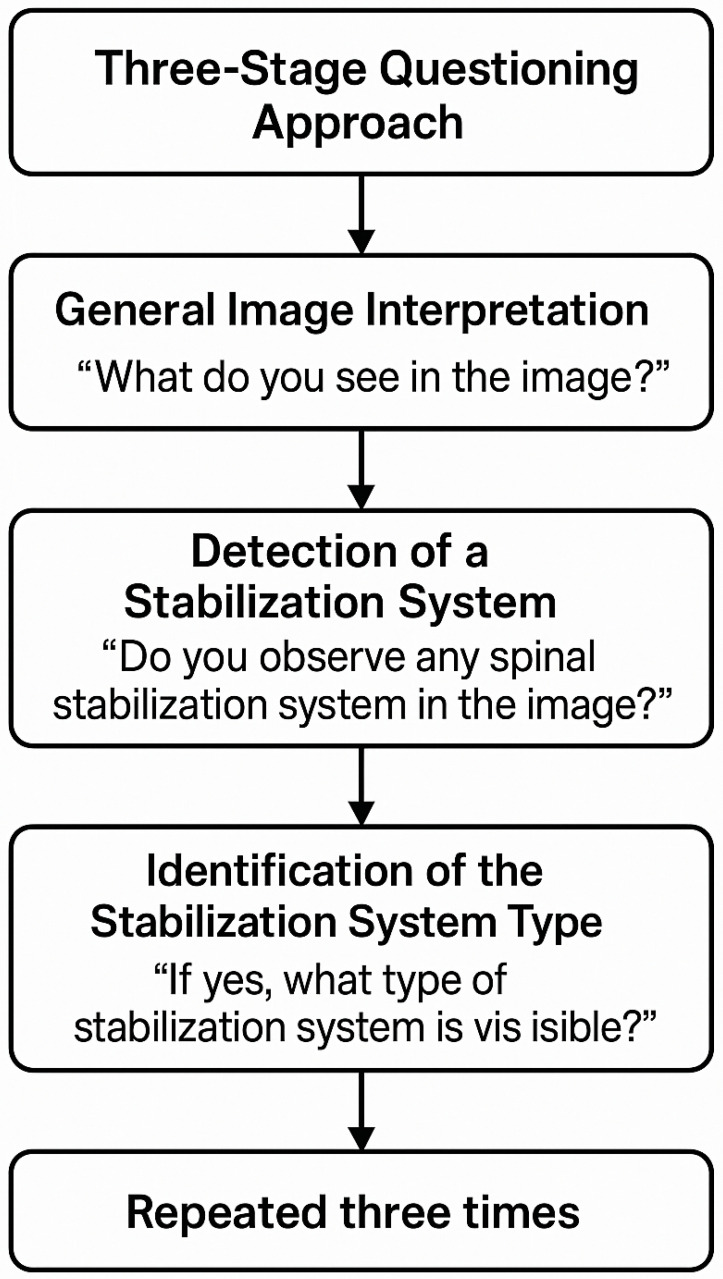
Flowchart illustrating the standardized three-stage questioning approach used to evaluate AI model performance. For each X-ray image, the model was asked: (1) to describe the image freely, (2) to determine the presence of spinal stabilization hardware, and (3) to identify the type of system if applicable. This sequence was repeated across three independent trials to assess consistency and diagnostic accuracy.

**Figure 3 jcm-14-03282-f003:**
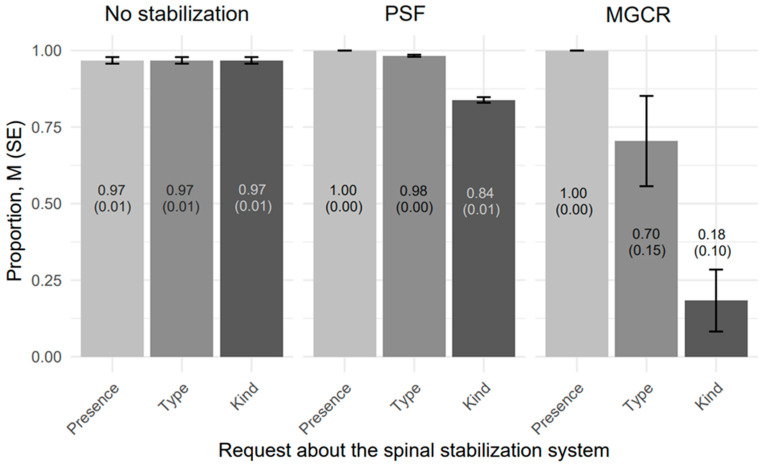
Comparison of detection proportions (here it is also a sensitivity level), M (SE), for GPT-4o’s in advanced scoliosis (Cobb angle ≥ 40°) using posturographic radiographic images.

**Figure 4 jcm-14-03282-f004:**
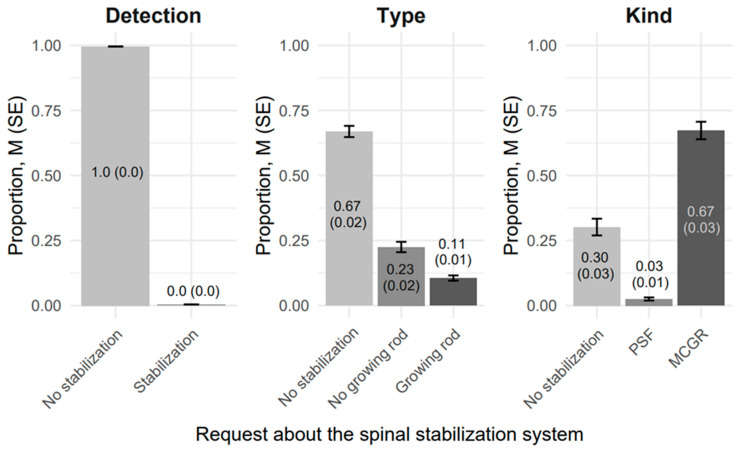
Comparison of detection proportions, M (SE), by BiomedClip AI in advanced scoliosis (Cobb Angle ≥ 40°) using posturographic radiographic images without stabilization systems (N = 93).

**Figure 5 jcm-14-03282-f005:**
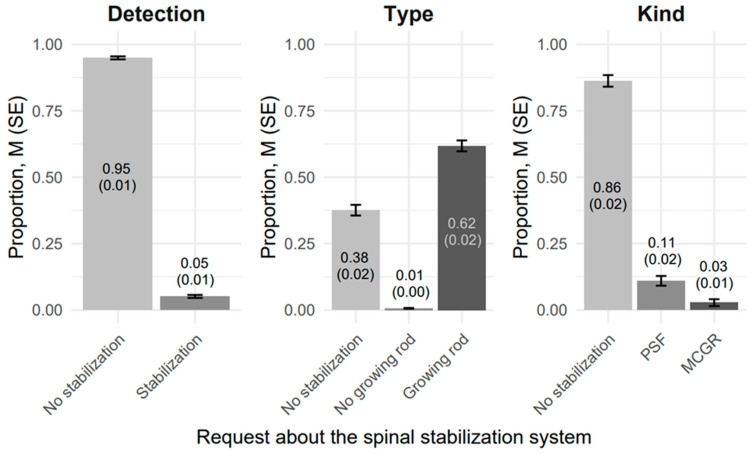
Comparison of the detection proportions, M (SE), by BiomedClip AI in advanced scoliosis (Cobb angle ≥ 40°) using posturographic radiographic images with PSF No-growing-rod stabilization systems (N = 97).

**Figure 6 jcm-14-03282-f006:**
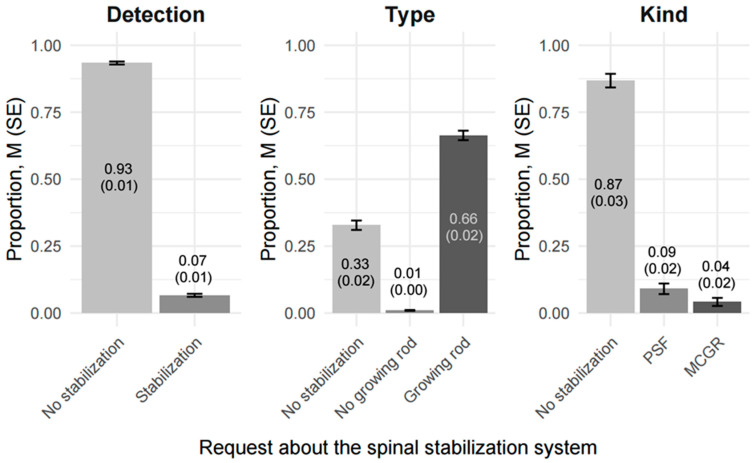
Comparison of detection proportions, M (SE), by BiomedClip AI in advanced scoliosis (Cobb angle ≥ 40°) using posturographic radiographic images with MCGR growing rod stabilization systems (N = 80).

**Table 1 jcm-14-03282-t001:** Hyperparameters used for fine-tuning BioMed CLIP, following the same setup as the original CLIP ViT-B/16 model.

Hyperparameter	Value
Learning Rate	4 × 10^−4^
Weight Decay	0.2
Adam β1	0.9
Adam β2	0.98
Adam ε	1.00 × 10^−6^
Batch size	32,768
Training epochs	32
Warm-up iterations	2000
Gradient clipping	Max temperature of 100

**Table 2 jcm-14-03282-t002:** Comparative sensitivity and inter-rater agreement for GPT-4o and BiomedCLIP in detecting spinal stabilization systems (SSS) in advanced scoliosis (≥40°) using posturographic radiographic images.

SSS Type	Classification Task	GPT-4o Sensitivity, % (95% CI)	BiomedCLIP Sensitivity, % (95% CI)	GPT-4o Gwet’s AC1 (95% CI)
**No Stabilization**	Presence	97.0 (94.0–100.0)	99.6 (99.6–99.6)	0.97 (0.94–1.00)
	Type	97.0 (94.0–100.0)	66.9 (63.0–70.8)	0.97 (0.94–1.00)
	Kind	97.0 (94.0–100.0)	30.2 (23.9–36.5)	0.97 (0.94–1.00)
**PSF**	Presence	100.0 (100.0–100.0)	5.1 (3.9–6.3)	1.00 (1.00–1.00)
	Type (Non-growing)	98.0 (96.0–100.0)	0.6 (0.4–0.8)	0.99 (0.97–1.00)
	Kind (PSF)	84.0 (78.8–89.2)	11.0 (7.8–14.2)	0.87 (0.80–0.94)
**MCGR**	Presence	100.0 (100.0–100.0)	6.6 (5.5–7.7)	1.00 (1.00–1.00)
	Type (Growing)	70.0 (51.2–88.8)	66.3 (62.8–69.8)	0.32 (0.16–0.49)
	Kind (MCGR)	18.0 (4.6–31.4)	4.2 (0.9–7.5)	0.50 (0.36–0.64)

Notes: Gwet’s AC1 for BiomedCLIP is 1.00 (perfect agreement) for all tasks and is not tabulated to avoid redundancy but is noted in the caption. Individual rater percentages and detailed agreement statistics (pa, pe, SE, *p*-values) are provided in Appendix A.

**Table 3 jcm-14-03282-t003:** Summary of comparative performance of GPT-4o and BiomedCLIP in detecting and classifying spinal stabilization systems, aligned with research hypotheses.

Hypothesis	Task	GPT-4o Sensitivity, % (95% CI)	BiomedCLIP Sensitivity, % (95% CI)	Sensitivity Difference (z, *p*-Value)	Cohen’s h	GPT-4o AC1 (95% CI)	Key Finding and Conclusion
**H1: Accuracy in Detecting SSS**	Presence (No Stabilization)	97.0 (94.0–100.0)	99.6 (99.6–99.6)	−2.6% (z = 2.7, *p* = 0.007)	−0.21	0.97 (0.94–1.00)	GPT-4o slightly less sensitive (small effect); both highly accurate, clinically negligible difference.
	Presence (PSF)	100.0 (100.0–100.0)	5.1 (3.9–6.3)	94.9% (*z* = 95.0, *p* < 0.001)	2.05	1.00 (1.00–1.00)	GPT-4o far superior (large effect), critical for clinical detection of PSF.
	Presence (MCGR)	100.0 (100.0–100.0)	6.6 (5.5–7.7)	93.4% (*z* = 93.0, *p* < 0.001)	1.95	1.00 (1.00–1.00)	GPT-4o highly reliable (large effect), BiomedCLIP inadequate for MCGR detection.
**H2: Consistency of Responses**	Type (No Stabilization)	97.0 (94.0–100.0)	66.9 (63.0–70.8)	30.1% (*z* = 13.44, *p* < 0.001)	0.68	0.97 (0.94–1.00)	BiomedCLIP perfectly consistent (AC1 = 1.00), GPT-4o variable but high agreement (medium effect).
	Type (PSF)	98.0 (96.0–100.0)	0.6 (0.4–0.8)	97.4% (*z* = 37.5, *p* < 0.001)	2.41	0.99 (0.97–1.00)	BiomedCLIP consistent but inaccurate, GPT-4o less consistent but accurate (large effect).
	Type (MCGR)	70.0 (51.2–88.8)	66.3 (62.8–69.8)	3.7% (*z* = 0.27, *p* = 0.790)	0.08	0.32 (0.16–0.49)	BiomedCLIP consistent, GPT-4o highly variable (low AC1, small effect).
**H3: Classifying Type/Kind**	Kind (PSF)	84.0 (78.8–89.2)	11.0 (7.8–14.2)	73.0% (*z* = 36.5, *p* < 0.001)	1.18	0.87 (0.80–0.94)	GPT-4o more accurate but less consistent (large effect), BiomedCLIP struggles with specificity.
	Kind (MCGR)	18.0 (4.6–31.4)	4.2 (0.9–7.5)	13.8% (*z* = 1.40, *p* = 0.160)	0.43	0.50 (0.36–0.64)	Both models suboptimal, GPT-4o slightly better but inconsistent (small to medium effect).

Notes: Sensitivity and 95% CIs are derived from Table 2. Sensitivity differences and z-tests are from the Statistical Analysis section. Cohen’s h is calculated as h = 2 × (arcsin
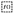
(√(p_1))-arcsin(√(p_2)), where p1 and p2are GPT-4o and BiomedCLIP sensitivities, respectively. Interpretation: small (h = 0.2), medium (h = 0.5), large (h = 0.8). GPT-4o AC1 values are from Table 2; BiomedCLIP AC1 = 1.00 for all tasks (noted in text). Conclusions summarize clinical implications, e.g., GPT-4o’s superior detection sensitivity (large effect sizes for PSF/MCGR) supports its use for initial screening, while BiomedCLIP’s consistency reveals the potential for structured classification with further optimization.

## Data Availability

The data are contained within the article.

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
