# Peer review of "Comparative Evaluation of Large Language and Multimodal Models in Detecting Spinal Stabilization Systems on X-Ray Images"

_jcm, 2025, doi:10.3390/jcm14103282_

Round 1

Reviewer 1 Report

Comments and Suggestions for Authors

The manuscript addresses a novel and highly relevant topic: the comparative use of language (ChatGPT-4o) and multimodal (BiomedCLIP) AI models to identify spinal stabilization systems in radiographic images. The methodology is carefully developed, and the article includes robust statistics. However, there are important areas that need to be strengthened before publication:

The hypotheses are poorly formulated and do not follow a robust scientific structure. For example, H1, H2, and H3 are worded as weak or redundant statements. Therefore, it is recommended to reformulate them following a clear relationship structure between the independent and dependent variables.

Although the GPT-4o evaluation process is well described, a precise description of how the model was interacted with, such as the interface, temperature, exact prompts, and model versions, is lacking.

The results table is extensive but not well aligned with the conclusions. Some interpretations are based on small differences without real clinical relevance. To improve its readability, it is important to add confidence intervals and effect sizes alongside the sensitivity comparisons.

Include a flowchart illustrating how the evaluation was conducted, especially the three-question outline for each image.

The poor performance of BiomedCLIP is not sufficiently explored. The model's specialization is mentioned, but potential training biases and the lack of specific fine-tuning for images with metallic implants are not discussed.

The language is sometimes redundant, and some sentences are poorly written or have unnecessarily complex syntax.

It would be advisable to add a section on the study's limitations.

Comments on the Quality of English Language

The quality of English is correct.

Reviewer 2 Report

Comments and Suggestions for Authors
  1. What specific criteria were used to determine the presence or absence of spinal stabilization systems in the X-ray images analyzed by the neurosurgeons?
  2. Could you elaborate on the methodology behind the three-stage AI-based questioning approach employed in this study for evaluating the performance of ChatGPT-4o and BiomedCLIP?
  3. How was the inter-rater reliability measured using AC1, and what does the range of AC1 values indicate about the consistency of the models' responses?
  4. What are the implications of the sensitivity analysis results, particularly regarding the stability of GPT-4o in hierarchical classification tasks and the performance deterioration of BiomedCLIP with increasing classification complexity?
  5. What are the potential clinical implications of the superior stability of GPT-4o in AI-assisted diagnostics for spinal stabilization systems?
  6. What steps could be taken to optimize BiomedCLIP's precision for complex radiographic evaluations, and what specific challenges does it face in recognizing PSF and MCGR configurations?
  7. Could you provide more details on the dataset of 270 X-ray images, including the criteria for inclusion and the demographic characteristics of the patients whose images were used?
  8. How does the study address the potential biases or limitations that might arise from using a dataset of a specific size and composition, and what measures were taken to ensure the generalizability of the findings?
  9. What are the potential limitations of using open-source AI models in the clinical setting for detecting spinal stabilization systems, and how do these findings contribute to addressing those limitations?
  10. It is important to understand how to further validate and improve the performance of AI models in the detection and classification of spinal stabilization systems on X-ray images.

Round 2

Reviewer 2 Report

Comments and Suggestions for Authors

Thanks for your revision.